# The SmartLandMaps Approach for Participatory Land Rights Mapping

Claudia Lindner [1,2], Auriol Degbelo [3,*], Gergely Vassányi [1], Kaspar Kundert [1] and Angela Schwering [1]

1    Institute for Geoinformatics, University of Münster, 48149 Munster, Germany;
     claudia.stoecker@uni-muenster.de or claudia.lindner@kadaster.nl (C.L.); vassanyigergely@gmail.com (G.V.);
     kasparjkundert@gmail.com (K.K.); schwering@uni-muenster.de (A.S.)
2    Kadaster, The Netherlands Cadastre, Land Registry and Mapping Agency,
     7311 KZ Apeldoorn, The Netherlands
3    Chair of Geoinformatics, Technische Universität Dresden, 01069 Dresden, Germany
*    Correspondence: auriol.degbelo@tu-dresden.de

**Abstract:** Millions of formal and informal land rights are still undocumented worldwide and there is a need for scalable techniques to facilitate that documentation. In this context, sketch mapping based on printed high-resolution satellite or aerial imagery is being promoted as a fit-for-purpose land administration method and can be seen as a promising way to collect cadastral and land use information with the community in a rapid and cost-effective manner. The main disadvantage of paper-based mapping is the need for digitization to facilitate the integration with existing land administration information systems and the sustainable use of the data. Currently, this digitization is mostly done manually, which is time-consuming and error-prone. This article presents the SmartLandMaps approach to land rights mapping and digitization to address this gap. The recording involves the use of sketches during participatory mapping activities to delineate parcel boundaries, and the use of mobile phones to collect attribute information about spatial units and land rights holders. The digitization involves the use of photogrammetric techniques to derive a digital representation from the annotated paper maps, and the use of computer vision techniques to automate the extraction of parcel boundaries and stickers from raster maps. The approach was deployed in four scenarios across Africa, revealing its simplicity, versatility, efficiency, and cost-effectiveness. It can be regarded as a scalable alternative to traditional paper-based participatory land rights mapping.

**Keywords:** land administration; paper map digitization; cadastral boundary extraction; vectorization; sketch maps; fit-for-purpose; participatory mapping; open data kit (ODK)

## 1. Introduction

Land rights mapping helps to establish and document secure land tenure, ensuring that individuals and communities have the legal recognition and protection of their rights to use, occupy, and benefit from land. It provides a clear record of land ownership, boundaries, and associated rights, reducing the risk of land disputes, encroachments, and forced evictions.

There are several innovative tools and technologies that can be used for land rights mapping. These tools leverage advancements in remote sensing, geographic information systems (GIS), and data analysis to improve the accuracy, reliability, efficiency, and transparency of land rights mapping processes [1,2]. Examples of innovative tools and technologies are manifold and include high-resolution satellite imagery and aerial imagery to capture detailed information about land cover, land use, and boundaries [3]; Unmanned Aerial Vehicles (UAVs) providing a flexible, cost-effective and timely collection of imagery of small to medium-sized areas [4]; GIS software providing a powerful platform for inte-

grating, analyzing, and visualizing land-related data; and mobile applications allowing for GNSS-supported field data collection and mapping.

Currently, millions of land rights worldwide remain digitally undocumented. Walking the boundaries of each owned property to record their spatial extent would not only be time-consuming but also impractical in some cases (e.g., huge plantations, swamps, dangerous areas). Hence, the challenge of documenting all land rights worldwide cannot be met with a classical surveying approach alone but requires complementary approaches. Participatory mapping using sketches presents an opportunity in this context. It can not only speed up the data collection process but also help capture local spatial knowledge from stakeholders and increase the community's confidence in the mapped information [5].

There are at least three approaches used to record spatial information via the use of sketches: *digital-sketching* (delineating the boundaries of a geographic entity is done with the help of a digital map, see, e.g., Refs. [6,7]); *analog-freehand-sketching* (the sketch is produced freehand on paper and no background map is used during the sketching, see, e.g., Ref. [8]); and *analog-sketching-on-map* (sketching is done over a georeferenced paper map, usually an aerial orthophoto, see, e.g., Ref. [9]). The three approaches have both advantages and disadvantages. As for digital sketching, the advantages include the efficiency of the data processing (i.e., the data is recorded digitally and can be automatically post-processed, analyzed, and combined with other datasets). The disadvantages are the learning curve (i.e., there is a need to teach people how to manipulate digital maps) and most importantly the logistics (i.e., either a whole community needs to be moved to the location of the Maptable [7,10] used to record the boundaries, or the Maptable needs to be transported to different locations). The key advantage of drawing on paper maps is that the logistics are much easier to cope with (see, e.g., Ref. [11]). Besides, it removes the technical hurdles of recording data, which means that the participants can focus entirely on the discussion about their surroundings and the matter at hand, rather than focusing on their interaction with a computer [11]. Finally, drawing on paper maps requires fewer instructions for the participants [11], and having a gentle learning curve for the participatory mapping activities is desirable so as to 'leave no one behind'. The key disadvantage of drawing on paper maps is the need for digitization to bring the data into a digital format. Currently, this digitization is mostly done manually, which is time-consuming and error-prone. The SmartLandMaps approach aims to address that gap.

The analog-freehand-sketching approach and the analog-sketching-on-map approach share the abovementioned advantages and disadvantages of drawing on paper. The main difference is that the use of aerial photographs or orthophotos as reference surfaces in the sketching-on-map strategy facilitates the preservation of spatial/geometric aspects of the drawn units. Since our goal in this work was to capture the outlines of spatial units, we followed a sketching-on-map strategy. The contexts of the four scenarios required the use of base maps from different sources (i.e., satellite and drone data providers) to ensure the most appropriate level of spatial detail and timeliness of the data during the participatory mapping activities. The focus of this work is on the following research questions:

- RQ1: How to facilitate scalable land rights recording?
- RQ2: How to automatically extract parcel boundaries from hand-drawn sketches?
- RQ3: How to automatically extract labels from hand-drawn sketches?

RQ1 is addressed through a participatory mapping-based strategy while RQ2+RQ3 are addressed through the SmartLandMaps software for the automatic extraction of parcel boundaries and parcel labels from paper maps. The contributions of this work are lessons learned from deploying the approach in four scenarios (RQ1) and evaluating the digitization software (RQ2+RQ3). As discussed in Ref. [12], cadastral boundaries can be broadly divided into two categories: (i) fixed boundaries, whose accurate spatial position has been recorded and agreed upon and (ii) general boundaries, whose precise spatial position is undetermined. The approach proposed in this work assists the first-time data collection and digitization of *general* boundaries.

The remainder of the article is structured as follows. Section 2 briefly presents related work on fit-for-purpose land administration, community-based mapping and the detection of boundaries from aerial imagery. Section 3 presents the SmartLandMaps approach, with a focus on preparation, the collection of informed consent, the mapping, and the processing of the collected data. The approach was deployed in four scenarios. Section 4 provides information about the study areas while Section 5 presents the evaluation results. Section 6 revisits the research questions and Section 7 concludes the article. This paper reuses and extends material from Ref. [13,14].

## 2. Related Work

### 2.1. Fit-for-Purpose Land Administration

Fit-for-purpose land administration (FFPLA) aims to close the global tenure security gap by providing secure and sustainable land rights to all members of society, in particular to those in informal or customary land tenure situations. In contrast to traditional (classical) survey approaches, it emphasizes the need for practical, affordable, and scalable solutions that can be implemented in a context-specific manner to meet the diverse needs and realities of different communities. Over the past decade, from its first publication in 2014 [15] and the establishment of basic principles that address spatial, legal, and institutional frameworks, the FFPLA has evolved into a viable concept that is implemented in various contexts [16]. Mapping visual boundaries using high-resolution satellite or aerial imagery is a key element of FFPLA methods for spatial data collection [15] and has been piloted across the globe [17–21].

### 2.2. Community-Based Mapping

Community-based mapping is a process of visualizing and understanding the physical, social, and economic characteristics of a community through the creation of maps. It involves collecting and analyzing information about the people, places, and resources within a particular community, and then presenting that information in a spatial format. The goal of community mapping is to create a comprehensive understanding of a community's assets, needs, and resources, and to use this information to inform community development, planning, and decision-making processes. Among others, community mapping has the following key qualities:

- Increased spatial knowledge: Community members have a deep understanding of their local area and can provide valuable insights and information that may not be reflected in traditional maps;
- Empowerment [22]: Community-based mapping empowers local residents to take ownership of their geographic information and to share their knowledge and perspectives with others;
- Increased local engagement: By involving local residents in the mapping process, community-based mapping can promote community engagement and strengthen local networks;
- Improved decision-making: Community-based maps can inform local decision-making and planning processes, ensuring that the perspectives and needs of local residents are considered.

Best practices in community engagement and various methods of participatory mapping have a long history in documenting land use and tenure and are used by various, mainly non-governmental organizations (e.g., Community Land Protection Facilitator Guide [23]). However, community mapping approaches have also gained importance in large-scale land tenure documentation projects (e.g., Rwanda) and are promoted as an FFPLA methodology [16,24]. Different tools and approaches can be utilized depending on the context, resources, and capacities [5,8,9]. However, in the realm of land administration, Ho et al. [25] revealed that while technology aims to promote inclusivity, it often falls short due to the lack of proper checks and the failure to view communities as equal partners in knowledge creation.

*2.3. Detection of Boundaries Using Machine Learning and Deep Learning*

Previous work has investigated the automatic detection of boundaries from aerial and satellite imagery. These boundaries can be either from buildings [26–31] or cadastral boundaries [32–38]. Since the focus of this work is on the automatic extraction of cadastral boundaries, we will now briefly review previous work touching on this topic.

Mango et al. [35] used neural networks to facilitate the process of converting paper-based cadastral maps into digital data. L-CCN was used to detect lines in Ref. [39], and ResNet-50 was used to detect numbers in Ref. [40] with promising results. Fetai et al. [34] used both the U-Net model [41] (open-source) and the ENVINet5 model (proprietary) while training deep neural networks on the task of automatic recognition of visible land boundaries. The areas selected for testing featured agricultural fields, roads, fences, hedges and tree groups. They reported accuracies greater than 95% for both models. Crommelinck et al. [36] used gPb (globalized probability of boundary) to automatically detect contours from orthoimages that show visible cadastral boundaries. They reported errors of omission between 14% and 52%. Crommelinck et al. [32] compared random forest (RF) and convolutional neural networks (CNNs) for the detection of cadastral boundaries and reported accuracies of 41% and 52%, as well as precisions of 49% and 76% for the two methods, respectively. Xia et al. [37] tested the performance of CNNs against MRS (multi-resolution segmentation) and gPb (globalized probability of boundary) for cadastral boundary detection in urban and semi-urban areas. They reported that CNNs outperformed MRS and gPb. The average quality assessment values obtained in their work for the CNNs were 0.79 in precision, 0.37 in recall, and 0.50 in F-measure. Finally, Persello et al. [38] used the SegNet model [42] to learn about the boundaries of agricultural fields in smallholder farms. They reported F-measures higher than 0.60 in their test areas.

Overall, many of the works presented above have relied on deep learning with promising results. Although deep learning models can learn complex characteristics that are challenging to specify manually, one drawback of deep learning methods is that they need large datasets for training. This is not the case in this work, where we only have a small number of instances (see Section 4). By contrast, conventional approaches (e.g., region-based, edge-based, and clustering-based) to image segmentation require less data. Their drawback, however, is the sensitivity to contrasts between objects and the background, and the subjectivity of the parameter selection (see Ref. [43]). We present an edge-based processing pipeline for boundary extraction in Section 3.4. We also report on the performance of the U-Net model [41] and the SegFormer model [44] on the boundary detection task. These two models have proven useful in many scenarios and correspond to the state of the art on semantic image segmentation. The U-net model was trained from scratch using a standard architecture. We fine-tuned pretrained SegFormer models on the sketches collected during the work, in the spirit of few-shot learning [45]. Image segmentation approaches using deep learning were reviewed in Ref. [46]. For a recent review of semantic segmentation in the context of geospatial artificial intelligence, see Ref. [43].

## 3. Method—The SmartLandMaps Approach

We build upon best practices for community engagement and participatory mapping methods while designing our mapping strategy (Figure 1). The mapping strategy is introduced in this section, with a focus on preparations (Section 3.1), the collection of informed consent (Section 3.2), mapping (Section 3.3), and the processing and digitization of the collected data (Section 3.4). For a discussion of the three pillars of the SmartLandMaps approach (acceptance, efficiency, and flexibility), see Ref. [47].

The recommended mapping process can be divided into three main phases: the preparation phase, the mapping phase, and the processing phase. The preparation phase should start at least six weeks before the mapping phase to ensure enough time for raising the community's awareness and allow for technical preparations. The actual mapping phase relies on a strong commitment from the local community and includes mobilization, an introduction to the mapping activity, obtaining informed consent and the actual collection

of spatial and textual data on land ownership and land use. The processing phase starts by tracing validated lines with a black marker, followed by taking photos of the map, which are then uploaded to the SmartLandMaps cloud along with the collected textual information. From here, land data can be fed into a national land administration system and further used for validation processes and issuance of land titles. The entire process requires only a tablet computer or smartphone, no software, and no sophisticated technical skills on the part of the community mappers.

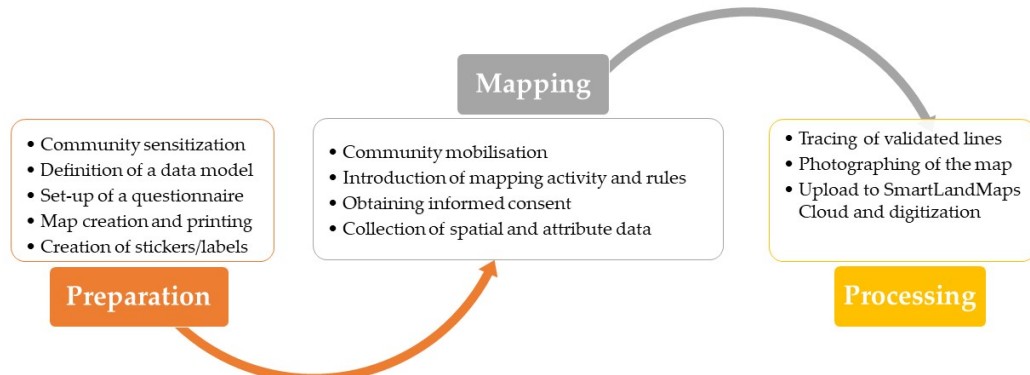

**Figure 1.** Mapping strategy of the SmartLandMaps approach.

*3.1. Preparations*

Preparations for community mapping processes are highly important and can significantly influence the success, accuracy, reliability, and effectiveness of the mapping initiative. Several aspects need to be considered to ensure that the process is well-organized, community-driven, and results in accurate, relevant, and impactful mapping outcomes. Table 1 outlines main activities and considerations in chronological order.

The purpose and scope of the community mapping activity should be at the forefront of any further decisions about the mapping process and subsequent preparation. Eight to four weeks before the planned fieldwork, a fieldwork plan should be drawn up. This will include decisions on the extent of the mapping area, a thorough review of existing procedures and practices that may influence the mapping process, and a data model and data collection strategy—all in cooperation and consultation with local partners, i.e., government representatives or local organizations. Once the mission plan has been drawn up, special attention should be paid to sensitizing the community, involving local leaders and community elders and gaining their trust and involvementin the process. If community members understand the purpose and benefits of the mapping activity, they are more likely to participate actively, contribute valuable information and take ownership of the results. The data model and the data collection strategy are introduced and discussed with community representatives. The data model should be appropriate to the mapping activity and local circumstances. Co-designing with local representatives ensures compliance with standards while meeting specific needs.

In addition to planning and consultation with local stakeholders, map production plays an important role. The printed orthophoto is the key mapping instrument, and if the map content or the resolution does not match the scope and the mapping objective, it is unlikely that the mapping activity will be successful. In particular, the selection of a suitable orthorectified image is important, especially in urban areas with multi-storey buildings causing relief displacement. Poor-quality maps can make it difficult to identify properties and respective boundaries. In this regard, the size of boundary objects and parcels should guide a decision on the required spatial resolution. Customary lands with several hundreds of hectares require a different spatial resolution of the base map than small plots in urban surroundings. At the same time, the data source for map creation should match the given timeline and budget. As a starting point, Enemark et al. [48] suggests different scales for mapping applications depending on topography and land use.

**Table 1.** Preparation steps for community mapping with the SmartLandMaps approach.

| What | When | Who | Comments |
|---|---|---|---|
| Definition of the objective of the community mapping activity | 8 weeks ahead | All parties involved | What to achieve? Whom to speak to? What challenge to solve? |
| Definition of the mapping area | 6 weeks ahead | Local partner | The area should be accessible, involvement of local representatives is crucial |
| Decide on and obtain the base data for the maps | 5 weeks ahead | Local partner (existing aerial data?), local drone company, satellite data provider | Based on local requirements, financial resources, data availability |
| Identification of current surveying practices, data models and local requirements | 5 weeks ahead | Local partner | Derive from country-specific land policies and survey manuals |
| Definition of a data collection process (and data model) | 4 weeks ahead | SmartLandMaps in consultation with local partners | Based on local characteristics/visual boundaries/mission objective/type of land tenure to be mapped |
| Creating a field mission plan | 4 weeks ahead | All parties involved | Decide on what needs to be done, by whom and with what kind of equipment |
| Sensitization of local stakeholders in the mapping area | 3 weeks ahead | Local partner | Share field mission plan and fine-tune requirements and activities based on local needs |
| Prepare and print base maps and mapping material | 2 weeks ahead | SmartLandMaps and local entity | Decide on layout, stickers and map features based on local circumstances |
| Circulate the final schedule of all activities | 1 week ahead | Local partner | Notify local representatives (and citizens) about meetings and activities |

### 3.2. Informed Consent

Following Bhutta [49], there are four determinants of the process of developing informed consent: (i) information provision and sharing by the research team with the participants and community leaders, (ii) discussion and interaction between researchers and potential participants, (iii) participant understanding, and (iv) acceptance/rejection of participation. To comply with these, we used an iterative model for consent. As discussed in Ref. [50], iterative models of consent are based on the assumption that ethical agreements can best be secured through a process of negotiation that aims to develop a shared understanding of what is involved at all stages of the research process. Hence, participants' agreement is not obtained through one-off (written) agreements, but the consent process is spread throughout the whole duration of the project (i.e., consent is asked on an as-needed basis at different stages of the data collection).

**Step 1: Sensitization and information (group consent)**: Before any data was collected, we informed all participants about the purpose, the procedure, the benefits, and the risks of participating in this research. We mentioned explicitly that the participation is voluntary and that the participants can quit the data collection activity at any time without having to state their reasons for doing so. Furthermore, we elaborated on the protection of any data collected and outlined the anonymization procedure. On site, a script detailing these aspects was developed and read out loud to the group. We then gave room for questions and gave the possibility for the attendants to leave the group if they did not want to participate. With this procedure, a first oral group consent was collected and the mapping and sketching could start with the persons manifesting their consent by staying.

**Step 2: Participatory mapping of the parcel boundaries**: The group that consented in step 1 discussed and marked the boundaries of their respective plots on the printed orthophoto provided by SmartLandMaps. Their discussions were not recorded. To protect

the anonymity of the participants, we took pictures without participants' faces (i.e., hands drawing on the paper sheet), photographing only the hands of those who consented. In this step, no additional personal information is collected.

**Step 3: Collection of personal information (oral informed consent)**: After the discussion and sketching session, we collected information on land ownership. Prior to starting the digital questionnaire, we again explained that the data collection is voluntary, that they can withdraw their participation at any time, and that they have to orally consent to continue with the questionnaire. We recorded the oral informed consent with a voice recorder, which was integrated into our questionnaire. The questionnaire did not foresee any obligatory fields regarding personal data. Hence, the participants could choose which data they wanted to provide and skip as many questions as they wished[1].

The above procedure (steps 1 to 3) was followed in the Benin scenario because the data collection in Benin was for research purposes only. In Chad and Sierra Leone, a slightly different procedure was followed in steps 1 and 3, as the data collection was embedded in a project setting with its own requirements for informed consent. In all cases, however, consent was sought in the local language of the community members as part of the voluntary group consent process.

*3.3. Mapping*

The community mapping process involves introducing community leaders to the process, often led by a local trusted body such as an NGO. Mapping materials and technology are introduced, questions are answered, and a mapping plan is developed. Meeting places are chosen at known meeting points within the mapping area to ensure that everyone can reach them easily. Mapping rules are agreed upon with local stakeholders. In some cases, separate mapping sessions for men and women may be planned to ensure that everyone can attend and actively participate in the mapping activity. Conflict resolution measures should be put in place, such as field-based boundary validation using GNSS technology if the boundary cannot be clearly defined from the orthophoto alone. According to Ref. [48], the field adjudication and recording process has three main elements: the location of the land right to be enjoyed, the nature of the right, and the person holding the right. The field adjudication process was supervised by a trusted intermediary such as a village elder or community official. It should be noted, however, that no title documents were issued as the field data collection was for research and demonstration purposes only, with an emphasis on the participatory process and technical feasibility.

**Boundary data collection:** A mapping assistant leads the mapping activity. The mapping assistant could be a para surveyor or a trusted person who knows how the mapping process works and what particular attention should be paid to the cartographic requirements and communication during the participatory mapping activity. In any case, he or she must take a neutral position. During the process, community members can also take charge of the mapping under the supervision of the mapping assistant. Property owners use landmarks such as churches, intersections, roads, sacred places, or schools to orient themselves on printed maps. If necessary, the mapping assistant will support the identification of landmarks and visible boundaries. Corner points of the plot are then marked either with small sticky dots or with dots drawn on the map with a ballpoint pen. Additional questions might be asked to confirm the location of the parcel such as: "Is this tree on your neighbor's plot, or does it belong to your plot? How many houses are on your plot?". The mapping assistant then connects the points with the help of a ruler and a ballpoint pen so that an area is created. When the property owner is satisfied with this, a spatial ID is assigned and a label is attached to the parcel that was just determined.

**Administrative data collection:** We used an Android App to perform the data collection. The Android App (Figure 2) was customized from the ODK collect app[2]. We created all questionnaires for the field studies ourselves using ODK Build. After the offline data collection in the field, the data is sent to a server once an internet connection is available.

We have our own server, located currently in a cloud from the provider DigitalOcean[3]. We chose DigitalOcean because it was recommended by ODK Collect. It has proven to be reliable throughout the whole project (18 months of testing). The data can be downloaded as a CSV (Comma Separated Values) file from our own ODK server. There is also the possibility of accessing the data from an API (Application Programming Interface). We have written a code (Python) that does the conversion from CSV to JSON (JavaScript Object Notation). Since the boundaries are extracted as JSON data as well, this enables the merging of the boundaries and administrative data using the parcel IDs obtained from the sticker extraction process (Section 3.4).

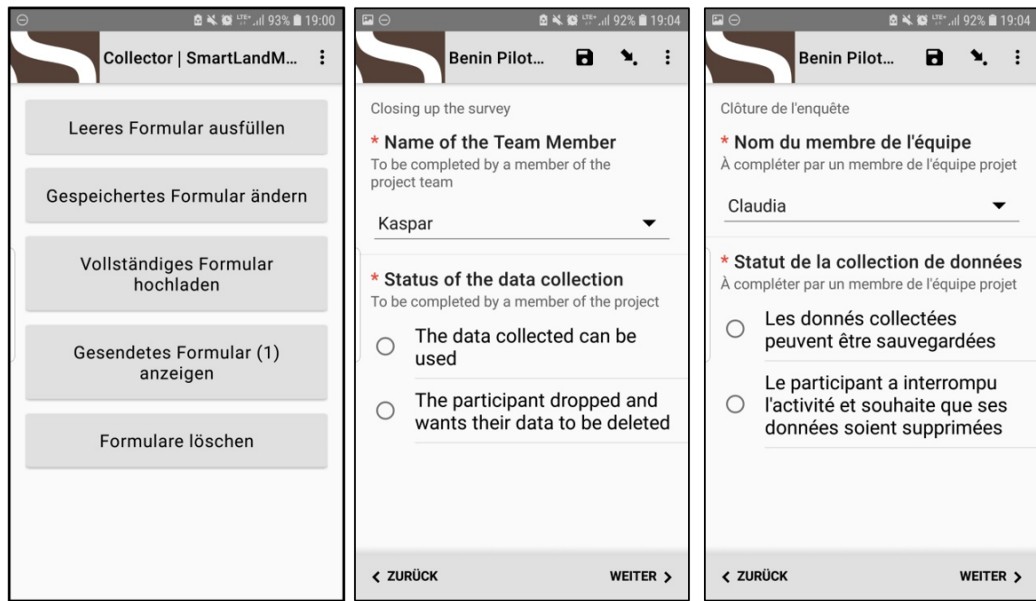

**Figure 2.** Screenshots of the Android App for data collection. Landing page of the app (**left**); an example of a question asked in English (**middle**); question translated in French (**right**).

### 3.4. Processing and Digitization

The processing pipeline involves several steps and algorithms. To convert the paper map to a digital format, the map is photographed with a consumer-grade camera, taking overlapping, non-tilted images. This allows the photos to be fed into a photogrammetric/structure-from-motion pipeline to produce a georeferenced, orthorectified image of the map. The creation of vector data from the georeferenced digital images is done by Python programs running from a Google Colab notebook. The boundaries are extracted from the images as binary rasters using computer vision algorithms and watershed segmentation. The binary raster is used as a basis for creating polygons, which are stored in a GeoJSON file. Another Python program deals with detecting printed numbers on the stickers, which are used for spatially joining attribute data to the vectorised parcels. The programs mentioned above are based on open-source libraries, which include OpenCV and skimage for image manipulation, gdal and pyproj for handling map projections and transformations, and shapely for constructing polygon geometries. For text recognition on the stickers, we used the Google Tesseract Optical Character Recognition Engine.

**Creation of a georeferenced orthorectified image:** The digitization starts with the processing of all photos taken from the sketched map using WebODM[4], with the aim of generating an orthophoto from the input data. The processing pipeline proceeds from motion structure, multi-view stereo, meshing, texturing, and ends with orthophoto generation [51]. As the photos of the map do not have usable geotags, the generated output is georeferenced using pre-existing markers (small red crosses) on the printed map. The location of at least five evenly distributed markers and the corresponding coordinates in the target reference system are used to apply the Helmert transformation. A georeferenced orthorectified

image of the annotated printed map is then generated and used for boundary detection and vectorization.

**Boundary detection:** The boundary detection is performed iteratively on patches of the original image, which are merged into a single binary image as an output result. The input image is first blurred using a 5 × 5 Gaussian filter, then it is converted into grayscale. These steps reduce noise and simplify the subsequent edge detection process. Edge detection is performed using the Canny algorithm, which detects significant changes in intensity. Next, Euclidean distance transformation is applied, which computes the distance of each pixel to the nearest edge, providing a measure of proximity to the boundaries. Using the distance-transformed image, peaks are detected to identify potential boundary locations. The watershed algorithm is applied to the negative of the distance-transformed image, utilizing the previously detected peaks as markers. Additionally, a binary mask is used to specify which regions of the image should be labeled. This binary mask is created by thresholding the grayscale image with a user-defined threshold value, which is one of the most important parameters. The threshold value was held constant at 60 for all techniques. The result is a segmented image with distinct labeled regions. Finally, all non-zero labeled regions are assigned a value of 255, while the background is assigned a value of 0, creating a binary image with the detected lines. In order to eliminate small gaps due to missed pixels close to patch edges and corners, this process is done again using a different patch arrangement. This time, padding is added to the image on the top and left sides, in a width equal to half the patch size. After the second binary image is obtained, the padding is cropped and the two results are combined using a bitwise OR operation. In the last step, dilation is applied using a 5 × 5 kernel to close the remaining small gaps in the boundaries. The steps described here share some similarities with the workflow presented in Ref. [52], but there are two key differences: no assumption that land parcel boundaries are straight lines and no use of the Hough transform during the process.

**Vectorisation:** The vectorisation of parcel boundaries starts with reading in the binary mask and applying a skeletonization algorithm, which shrinks all lines to a minimum width. Next, using OpenCV's findContours() function, the contours of the shrunk lines are detected and filtered so that only inner contours remain. This means that all gaps in parcel boundaries have to be closed in order to get a contour from the parcel. Using gdal.Info, georeferenced coordinates and coefficients are obtained from the original image, which are then applied to create georeferenced polygons. The script iterates over the contours and creates UTM polygons using Shapely's Polygon function. The polygons are transformed into WGS84 with pyproj, and are filtered by their geodetic area. We chose 50 m$^2$ as the lower threshold. Then, a GeoJSON FeatureCollection is created with two predefined attributes, parcelID (set to an integer starting from 1) and parcelType. The script permutes over the coordinates and creates a JSON-compatible geometry object for each polygon. The output is written into a GeoJSON file.

We have tested two different vector post-processing approaches in order to minimize the deviation between the generated polygon shapes and the original boundaries. Figure 3 illustrates the effect of both methods. The first method (Figure 3, left) creates a minimum area convex hull over the raw polygons by utilizing Shapely's convex_hull method. The program then removes overlapping areas and returns a new dictionary with all the features. This approach works generally well, eliminating small holes and producing straight polygon boundaries. The drawback of this method is that it cuts corners for individual concave polygons, therefore, it produces wrong boundaries. These polygons either need manual editing or have to be supplied to a fixing algorithm, which subtracts the largest polygon out of the difference between the pre-processed and post-processed geometries. The second approach (Figure 3, right) first filters the points of the raw polygons to only keep the exterior ring, which eliminates all possible holes. Generalization and cleaning are done by GRASS GIS commands, which are invoked from a GRASS session by utilizing the grass_session Python library. This requires GRASS to be installed on the engine where the script is running. After creating a custom Session instance, we open a

new mapset and import the raw polygons GeoJSON with the v.in_ogr command. A minimal snapping tolerance ($1 \times 10^{-10}$) is applied, which fixes some topological errors. Next, we perform Douglas–Peucker generalization by using v.generalize. The generalization threshold is adjustable and depends on the sizes of the parcels and we have decided to use $5 \times 10^{-6}$. The v.generalize command also creates polygons to fill small gaps that happened during the polygon creation. v.clean dissolves these small polygons into an adjacent one with the largest common boundary. v.out_ogr creates a GeoJSON output of the final result. Compared to the convex hull approach, GRASS retains a more accurate shape for the polygons, especially concave ones, but can fail to straighten the boundary at noise-related bends and errors. Preliminary testing [14] has shown that the optimal threshold for boundary extraction is dependent on the image. This is also the case for the optimal parameter for the generalization algorithm. Ideally, these parameters should be adapted based on the scenario. Nonetheless, this is not practical for comparison activities across all scenarios. Hence, we chose two values for the comparative assessment in this article based on preliminary tests: 60 as a threshold value for the boundary extraction and $5 \times 10^{-6}$ as a generalization threshold for the vectorization. We are aware that this inevitably flavors some methods in some conditions to the detriment of others.

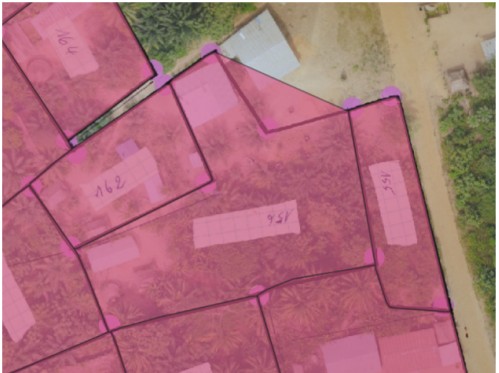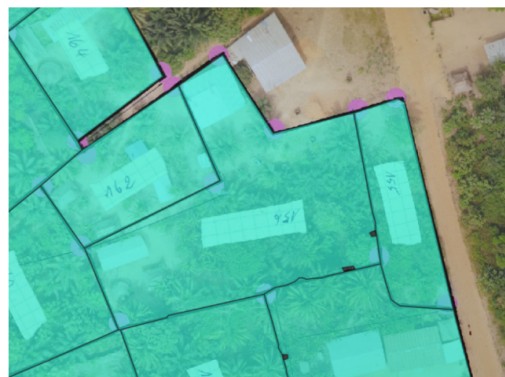

**Figure 3.** Examples of the convex hull method and the GRASS method, respectively.

**Sticker detection:** Sticker detection is performed by first filtering the RGB channels of the image so that the resulting binary image only contains the stickers as white pixels and everything else as black. For this thresholding, a minimum and a maximum value (0–255) must be specified for all 3 channels as the input parameters. As we had bright yellow stickers, we used the following values: Red—between 160 and 255, Green—between 150 and 255, and Blue—between 0 and 120. After filtering, morphological closing and blurring are applied to close holes and reduce the noise. Next, OpenCV's findContours() function detects all contours and keeps only the ones above a size threshold. For each contour, an algorithm creates an approximate polygon, which is then used for determining a minimum area rotated rectangle. The coordinates of the central point are saved for the GeoJSON creation in a list. Optical Character Recognition requires numbers to be aligned horizontally. For this reason, the rotated rectangles are rotated back to horizontal, resulting in a clip of the original image, on which the numbers are either in the desired position or upside down. We use an asterisk as the last character of each text on the stickers (see Figure 4, left), which, if not detected, indicates that the sticker is upside down and the snippet needs to be flipped.

During the rotation process, a black padding is added to maintain a rectangle shape by using the OpenCV function imutils.rotate_bound. As we are only interested in the center of the snippets, contour detection and polygon creation are performed again, and the snippets are cropped by this polygon. The resulting image is now ready for OCR to be applied. To increase efficiency, detection is performed five times on different resized versions of the image, in the range where performance is best. A function analyses the text and returns an assumption parameter, whether the detection was successful or not. Based on the

five results, the program chooses the mode value from the ones labeled good (if any). The output GeoJSON file is a FeatureCollection containing all detected stickers as point features with the detected number and the assumption as parameters (Figure 4).

Original image

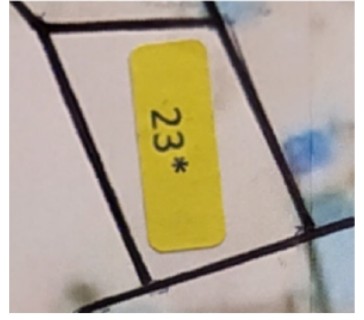

Sticker detection process →

GeoJSON Result

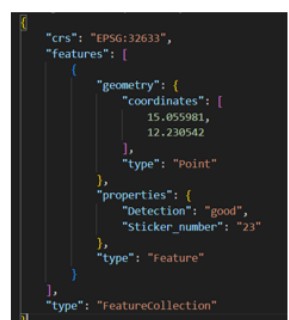

**Figure 4.** An example of the output from the sticker detection process.

## 4. Study Areas

The data input for this paper comes from four different scenarios, including projects in Benin, Chad, and Sierra Leone. The mapping activity was carried out by organizations partnering with SmartLandMaps in all three countries, each with its own unique history and challenges. In Benin, a centralized land administration system is being established, with mapping supported by the Dutch-funded Land Administration Modernisation Project [53]. Sierra Leone faces problems of land grabbing [54] and is working on land tenure security through novel land policy reforms and major donor funding through the World Bank's Land Administration Project. Chad is tackling challenges such as slow mapping and land conflicts, with support from the Dutch Ministry of Foreign Affairs' LAND-at-scale program. As described in Table 2, mapping activities in these countries have involved collaboration with local communities, government agencies, and technology providers.

**Table 2.** Characteristics of the study areas, taken from Ref. [14].

|  | **Benin** | **Chad** | **Sierra Leone** |
|---|---|---|---|
| *Partners involved* | Kadaster International, VNG International, YILAA | Kadaster International, Government of Chad, esri North Africa, Trimble, University of Twente | FIG YSN VCSP, Trimble, Ministry of Land, Housing and Country Planning |
| *Date* | 1–9 February 2022 | 11–12 October 2022 | 24–25 January 2023 |
| *Orthophoto* | UAV-based orthophoto, 1.8–2.3 cm | MAXAR, 50 cm | MAXAR, 50 cm |
| *Land use class* | Urban residential and rural residential | Peri-ruban residential | Rural agroforestry |

The land-use context, as well as the land tenure system and the size of the spatial units, were heterogeneous across the scenarios. In Benin, the data collection took place in an urban setting with private ownership (Seme-Podji) and in a rural setting with individual and group ownership (Zè) [13]. In Chad, SmartLandMaps was tested alongside other forms of cadastral surveying and mapping in a peri-urban environment near the capital N'Djamena, characterized by private property rights [55]. The context in rural Sierra Leone was very different, with predominantly agroforestry land use and a customary tenure system, i.e., family lands averaging several hundred hectares. Looking at Figure 5, it is clear that the presence of visible boundaries was different in each scenario, allowing different strategies to be observed in how community members deal with them. For more

information on the contextual specifications, data collection, observations and results, see Ref. [14].

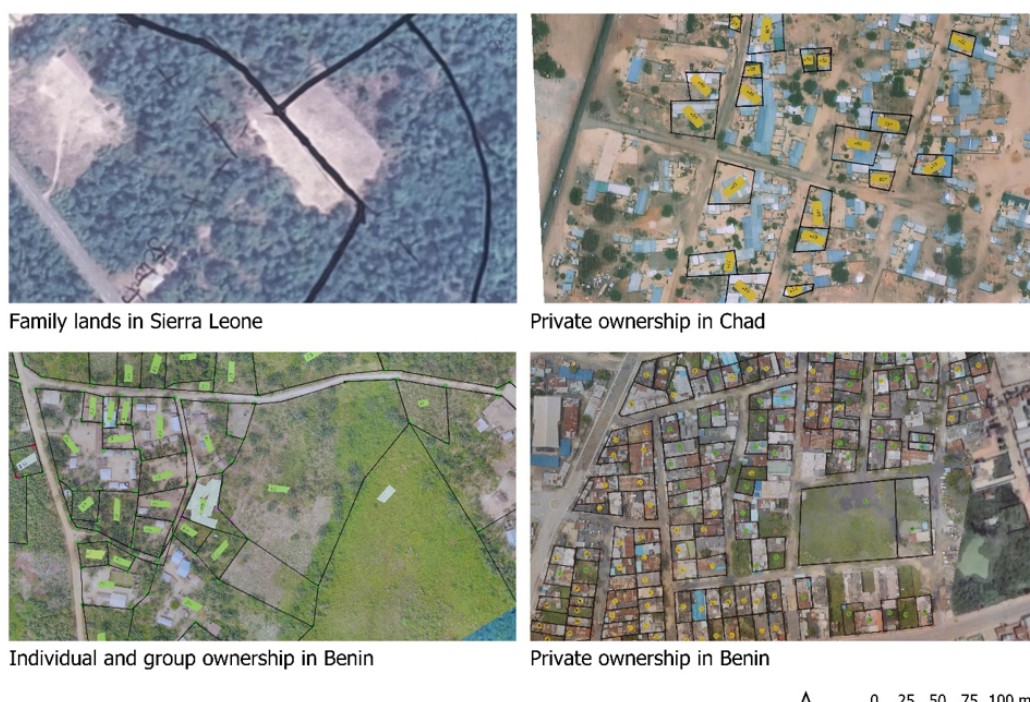

Figure 5. An excerpt of the community mapping outputs. The scale is the same for all maps.

## 5. Evaluation

The evaluation touches upon five aspects: the digitization performance, cost and time considerations, the simplicity of the mapping activity for participants, the inclusiveness of the whole approach, and its versatility.

### 5.1. Digitization Performance

The tests were done in two steps: first a comparison of the convex-hull-based approach and the Douglas–Peucker approach to find out the best between the two, and second an assessment of the impact of the pre-processing strategy (Blurring or Mean-Shift) on the performance of the best approach. Table 3 shows the results. Overall, the test on the four scenarios (Benin urban, Benin rural, Chad, and Sierra Leone) show that our method can achieve excellent performance on different datasets in a quick timeframe. Depending on the size of the input image, the boundary detection took about 3 min (Chad-North scenario) to 16 min (Benin rural scenario). The vectorization only took 1 to 3 min, similar to the sticker detection. Since the performance values were obtained without adjusting the threshold parameter (boundary extraction algorithm) and the generalization parameter (vectorization algorithm), the values in Table 3 should be seen as the *lower bounds* of the final performance. That is, calibration specific to a scenario can further increase them.

Table 3 also presents the performance results for model training (U-Net) and model fine-tuning (SegFormer model, nvidia/mit-b1) on the dataset. As discussed in Section 2.3, deep learning models are not entirely appropriate for the current context given the small amount of data points. Nonetheless, since U-Net and SegFormer stand for the state of the art in image segmentation, we trained the two models to use their results as a baseline for the comparison of the proposed algorithms. The dataset from the Benin urban scenario, which had the largest number of parcels (see Figure 5), was used for training. The outcome map was manually labeled using two classes (background and boundary) and then divided into patches (512, 512, 3). This resulted in a total of 1260 patches. About 70% of these

patches were used for training, 20% for validation, and 10% for testing. The dataset was not balanced (723 patches ~57% showed a boundary and 537 patches ~43% did not show any boundary). We did not use data augmentation. The metrics obtained on the validation dataset during training were: 99% (accuracy, U-Net); 48% (Mean IoU, U-Net); 92% (accuracy, SegFormer); 86% (Mean IoU, SegFormer). Examples of prediction outcomes per patch on the test dataset (i.e., unseen patches) are shown in Figure 6. We fine-tuned both nvidia/mit-b0 and nvidia/mit-b1. Since we obtained slightly better results with the nvidia/mit-b1 model, we only report the results for nvidia/mit-b1.

In the case of the Benin and Chad scenarios, blurring with Douglas–Peucker generalization (using GRASS) produced the best performance regarding boundary detection, with a 0.99 F-measure. We can see a slight improvement in both precision and recall compared to the Convex hull method. The most significant difference is observed with the Benin urban scenario, where the number of true positives exceeds the Convex hull result by 6, while the numbers of false positives and negatives are simultaneously lower. In all cases, there have been no invalid geometries using GRASS, as opposed to a few cases with the Convex hull. Both approaches led to a significant number of features that require manual editing, which suggests that both methods call for further improvements. Mean-Shift proved to be less efficient with lower recall values than Convex hull and Douglas-Blur for both Benin images. In the case of Chad, where all three methods produced the same F-measure, the number of features needing to be edited is almost double that of the other methods.

Results show a much lower efficiency for all the above-mentioned methods in the case of the Sierra Leone scenario. The cause of this issue is linked to the boundary detection threshold parameter, which we left unchanged for the sake of comparison. As this image is generally brighter, a higher threshold would have been necessary, as it is directly linked to the detection of lines. Using image correction techniques (such as brightness and contrast adjustment), this concern could be addressed. This also indicates that without raster pre-processing, choosing the optimal boundary parameter is essential.

Out of the two neural networks, U-Net produced far better results than the SegFormer model. It managed to obtain the best F-measure for Sierra Leone with 0.56 and yield 0.91 for the Chad scenario. Arguably, these results do not come close to our best-performing method with optimal parameters, but they show that U-Net could be a promising alternative in the future. Nevertheless, efficiency is vastly dependent on the training and testing data, and detecting boundaries over different backgrounds remains a challenge.

The Benin scenarios were excluded from the sticker detection, as they only contain stickers with handwritten digits, in a non-standardized form. In the case of the Chad scenario, there was only 1 missed sticker out of the 52 cases, which means 0.98 accuracy and complete precision. However, in the Sierra Leone scenario, none of the 11 stickers were detected. The reason behind this poor result is the same as mentioned above, namely the unchanged parameters for two images with different characteristics. The sticker detection can achieve great proficiency, but similarly to boundary vectorization, it relies on the appropriately chosen parameters unless raster pre-processing is involved.

### 5.2. Cost and Time Considerations

The costs and time needed for capturing boundary data determine, whether an approach used in a small pilot project is scalable, i.e., whether it could eventually be rolled out for larger regions or be used to create a country-wide cadaster. A cadastral project can be called "fit-for-purpose" if it is implementing a good compromise between the desired positional accuracy on the one hand, and the available resources in terms of money and time on the other hand. Our practical experience in the mentioned four scenarios showed that the SmartLandMaps approach is suitable for projects with comparatively low costs per parcel and time to complete a larger data capture exercise. In Benin and Chad, we were able to capture with one field team up to 200 parcels per day when using the SmartLandMaps approach, while the same team achieved only 20–30 parcels when walking to and measuring the boundary points of each parcel with a handheld device

and a GNSS antenna. The costs incurred for the data capture with the SmartLandMaps approach averaged from 5 to 10 USD/parcel including the drone image and the processing of the data in the SmartLandMaps Cloud, while the direct surveying with the handheld device we did for comparison resulted in costs from 20 to 50 USD/parcel. These costs are in line with the unit costs stipulated by UN-Habitat, FIG, and GLTN in their framework for costing and financing land administration services (CoFLAS [56]). From these observations on costs/time and the digitization performance (Section 5.1), we can state that the Smart-LandMaps approach is fit-for-purpose: It is reliable, faster, low cost (4–5 times cheaper), and scalable (adding more field teams does not necessitate much investment in training), but requires, as a compromise, that reachable positional accuracy is 0.8 to 1.5 m lower than the positional accuracy of traditional surveying.

**Table 3.** Digitization performance (TP: True Positives; FP: False Positives; FN: False Negatives; NG: Features with NULL geometry; NE: Needs to be edited; N/A: Not applicable). The highest metric for a scenario is highlighted in grey.

| | TP | FP | FN | NG | NE | Precision | Recall | F-Measure | Scenario |
|---|---|---|---|---|---|---|---|---|---|
| slm-Convex-hull | 191 | 4 | 10 | 4 | 30 | 0.98 | 0.95 | 0.96 | Benin urban |
| slm-Douglas-Blur | 197 | 1 | 4 | 0 | 33 | 0.99 | 0.98 | 0.99 | |
| slm-Douglas-Mean-Shift | 161 | 0 | 42 | 0 | 70 | 1.00 | 0.79 | 0.88 | |
| | | | | | | | | | |
| slm-Convex-hull | 74 | 0 | 3 | 1 | 8 | 1.00 | 0.96 | 0.98 | Benin rural |
| slm-Douglas-Blur | 75 | 0 | 2 | 0 | 8 | 1.00 | 0.97 | 0.99 | |
| slm-Douglas-Mean-Shift | 71 | 0 | 6 | 0 | 11 | 1.00 | 0.92 | 0.96 | |
| unet-Convex-Hull | 19 | 2 | 58 | 0 | 2 | 0.90 | 0.25 | 0.39 | |
| unet-Douglas | 19 | 2 | 58 | 0 | 2 | 0.90 | 0.25 | 0.39 | |
| segformer-Douglas | 8 | 0 | 69 | 0 | 2 | 1.00 | 0.10 | 0.19 | |
| segformer-Convex-Hull | 8 | 0 | 69 | 0 | 2 | 1.00 | 0.10 | 0.19 | |
| | | | | | | | | | |
| slm-Convex-hull | 51 | 0 | 1 | 1 | 9 | 1.00 | 0.98 | 0.99 | Chad north |
| slm-Douglas-Blur | 51 | 0 | 1 | 0 | 8 | 1.00 | 0.98 | 0.99 | |
| slm-Douglas-Mean-Shift | 52 | 1 | 0 | 0 | 15 | 0.98 | 1.00 | 0.99 | |
| unet-Convex-Hull | 43 | 0 | 9 | 0 | 5 | 1.00 | 0.83 | 0.91 | |
| unet-Douglas | 44 | 0 | 8 | 0 | 3 | 1.00 | 0.85 | 0.92 | |
| segformer-Douglas | 13 | 0 | 39 | 0 | 2 | 1.00 | 0.25 | 0.40 | |
| segformer-Convex-Hull | 13 | 0 | 39 | 0 | 3 | 1.00 | 0.25 | 0.40 | |
| | | | | | | | | | |
| slm-Convex-hull | 4 | 0 | 8 | 0 | 4 | 1.00 | 0.33 | 0.50 | Sierra Leone |
| slm-Douglas-Blur | 4 | 0 | 8 | 0 | 1 | 1.00 | 0.33 | 0.50 | |
| slm-Douglas-Mean-Shift | 1 | 0 | 11 | 0 | 0 | 1.00 | 0.08 | 0.15 | |
| unet-Convex-Hull | 5 | 1 | 7 | 0 | 3 | 0.83 | 0.42 | 0.56 | |
| unet-Douglas | 5 | 1 | 7 | 0 | 0 | 0.83 | 0.42 | 0.56 | |
| segformer-Douglas | 0 | 0 | 12 | 0 | 0 | 0.00 | 0.00 | 0.00 | |
| segformer-Convex-Hull | 0 | 0 | 12 | 0 | 0 | 0.00 | 0.00 | 0.00 | |
| | | | | | | | | | |
| Sticker detection | 0 | 0 | 0 | N/A | 0 | 0.00 | 0.00 | 0.00 | Sierra Leone |
| Sticker detection | 51 | 0 | 1 | N/A | 0 | 1.00 | 0.98 | 0.99 | Chad north |

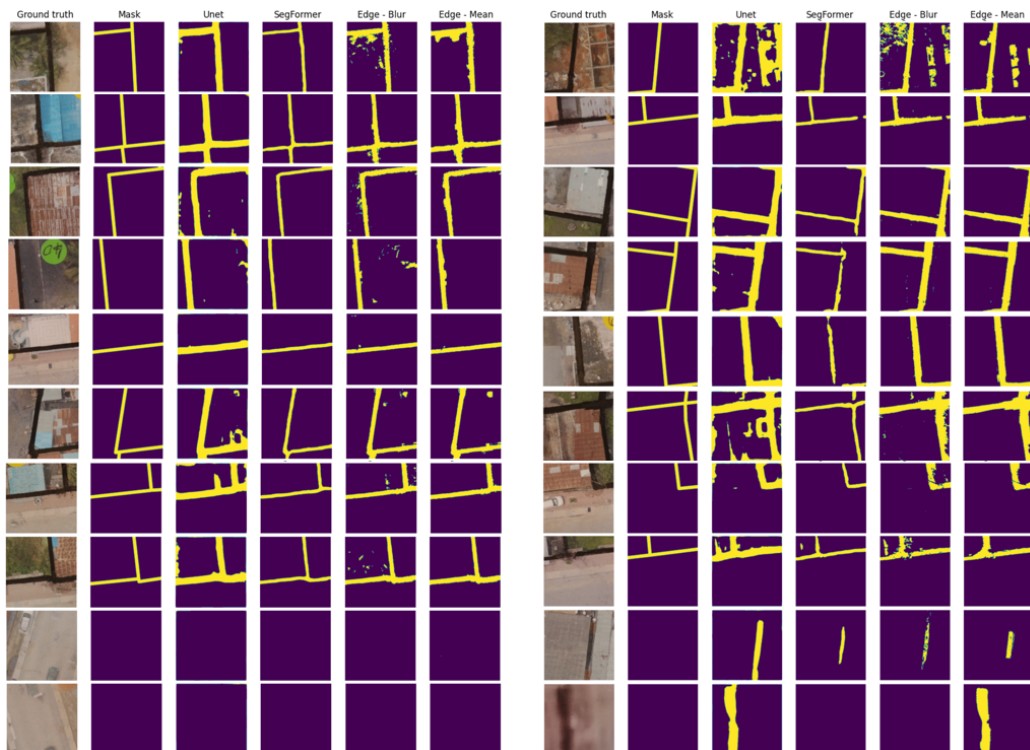

**Figure 6.** Examples of boundary predictions on the test dataset. Patches where the whole boundary is correctly predicted (**left**); patches where the boundary is not correctly predicted for at least one of the techniques (**right**). The boundaries predicted at the patch level are then merged (and post-processed to close gaps) to obtain the parcels for a whole study area.

### 5.3. Simplicity

The main mapping activity required only a pen, a printed orthophoto, and a mobile device with a camera. Because SmartLandMaps allows for an almost fully automated digitization workflow, mapping assistants need only very basic skills to facilitate the mapping activity, but also to initiate digitization. Simplicity applies not only to the mapping process, but also to the ease with which community members engage with the map as an accessible and easy-to-understand method of collecting spatial information. According to a survey in Benin ($n$ = 388), almost 90% of participants reported that they found it either very easy (65%) or fairly easy (22%) to mark the boundaries of their parcels on the map [13]. However, it is important to note that this result can be influenced by the type of land cover and land use. Adjustments to the ease of mapping may be necessary in cases of homogeneous land cover [57].

### 5.4. Inclusiveness

Inclusiveness can refer to the diversity of tenure systems or to the dimension of people involved. In the three study sites, SmartLandMaps proved to be inclusive of both formal and informal tenure systems. As for the dimension of inclusiveness of the mapping process, we observed all kinds of participants in the mapping session, including women and men, old and young, people with disabilities, educated and literate, as well as illiterate and less educated people. However, it should be noted that future data collection campaigns should consider measures to further increase women's participation in the mapping process, which was particularly low in Chad and Sierra Leone.

### 5.5. Versatility

It was shown that SmartLandMaps can be applied in different contexts with an adapted community mapping process, depending on different tenure systems, community structures, and visibility of boundaries. Where boundaries are poorly visible and the

participatory mapping process alone cannot produce reliable sketches on the orthophoto, a combination with additional ground measurements where necessary can be considered. Easily identifiable labels allow for the merging of non-spatial data with spatial units. In the future, different colors for the labels can even allow for different layers of information on one map.

## 6. Discussion

SmartLandMaps emphasizes the importance of community participation and co-creation of information, all while ensuring that the approach does not compromise on the need for digital data handling realized through a semi-automated digitization pipeline. Introducing the SmartLandMaps approach to the existing set of land tenure recording tools can be a significant step forward, especially when we consider the general benefits highlighted in Rambaldi's work on participatory GIS (PGIS) [58].

### 6.1. Key Takeaways

We now revisit the three research questions mentioned at the outset of the article and summarize the main lessons learned.

**How to facilitate scalable land rights recording?** A participatory mapping approach, combined with efficient digitization techniques can be useful for general boundary recording in a time-efficient manner. The digitization of the maps is fast (see Section 5.1), which means that the bottleneck of the approach is the effort needed to mobilize the participants and run several mapping campaigns in parallel. It should be noted that, even though the mapping sessions were carried out at different locations, contexts, and cultures, the printed orthophoto was always the key instrument for interaction, cohesion, and consensual spatial decision-making. In this sense, one could argue that the process was as important as the result, as observed in other studies as well [59]. Moreover, it was observed that SmartLandMaps has relatively low requirements when it comes to technological know-how, making it accessible and easily adopted by local entities. It only requires a minimal amount of training for local community members to become proficient in its use. In this vein, a train-the-trainers approach can easily be applied to keep the knowledge in the country [24].

**How to automatically extract parcel boundaries from hand-drawn sketches?** We have tested several techniques for boundary detection and boundary vectorization during the work. Table 3 shows good results for our digitization approach. Blurring as a preprocessing technique yields slight improvements in comparison to mean-shift for the detection task. For vectorization, Douglas–Peucker has often led to better results than the use of convex hulls. The two machine-learning models tested did not always yield performance as good as the algorithm for boundary extraction proposed. Since the prediction at the patch level was good (Figure 6), this is an indication that more post-processing is needed to close gaps between predicted boundaries at the patch level when reconstructing the final image. We defer a closer look into this to future work.

**How to automatically extract labels from hand-drawn sketches?** We have used a workflow including the printing of numbers on stickers that are placed on the paper map, a self-written script that uses OpenCV to search for stickers in the digitized map image and Tesseract for character recognition with promising results (Table 3). Early tests using hand-written digits without a symbol to mark the end of numbers, as shown in Figure 4, produced less reliable results.

### 6.2. Limitations

The digitization of cadastral boundaries introduces positional error as documented in Ref. [60]. In our case, these positional errors related to the boundaries may come from the different thicknesses of the lines drawn by participants and also the fact that their drawing may not perfectly align with the outlines of the boundaries on the ground. We

have documented an assessment of the positional accuracy of the approach in Ref. [13]. The SmartLandMaps approach is useful for first-time recording to get an inventory of the parcels available in a region, as well as their general outlines (i.e., the general-boundary approach). It can be followed by more rigorous surveying approaches (i.e., the fixed-boundary approach) to increase the accuracy of the positions of the boundaries in the digital land administration systems.

### 6.3. Future Work

The SmartLandMaps approach as discussed in this manuscript presents a novel procedure to easily digitize sketched and annotated maps. However, the process of community mapping can also disadvantage minorities and exacerbate social imbalances if special attention is not given to it from the outset. As this paper aimed at a proof of concept of the technology and mapping methodology, this aspect was not the subject of this investigation and could be taken up in subsequent studies, i.e., how to ensure that all individuals and not only elites participate in the mapping session. As for the digitization, the results have shown promising results and could be extended in at least three ways. First, edge detection relied primarily on canny edge detection in this work. It would be interesting to explore alternative edge detection mechanisms (e.g., L-CNN [39]) and their impact on the results. Second, the tested machine learning-based approaches have shown promise, despite the limited training data used. This suggests that, with more data and empirical testing, we may arrive at models with even better results than those observed in this study in the future. In particular, it would be interesting to explore how U-Net and SegFormer will perform with more training data and/or data augmentation, and how additional models for semantic segmentation (e.g., TransUnet [61], UnetFormer [62]) would contribute to the automatic extraction of cadastral boundaries in future work. Third, more data in additional scenarios would be useful to increase the diversity of the data collected and increase the generalizability of the results to various types of participatory mapping contexts.

### 7. Conclusions

Millions of formal and informal land rights are still undocumented worldwide and there is a need for scalable techniques to facilitate that documentation. Through the combined use of satellite or drone-based aerial photography, participatory mapping, and digitization software, the SmartLandMaps approach reduces the time and costs to create digital data with information on the extent of a land right (parcel boundaries), the land rights, restrictions and responsibilities, as well as information on the land right holder. This article has presented an overview of the different components of the approach and lessons learned from its deployment in four scenarios. The SmartLandMaps approach proved to be efficient, with digitization being a reliable and quick step, in contrast to previous workflows where the proper digitization of paper sketch maps was a major bottleneck and often prone to errors. The importance of community interaction and the mapping process itself was emphasized, with the printed orthophoto being a key tool for interaction and decision-making for all community members during the mapping sessions. The SmartLandMaps approach was also accessible and easily adopted by local entities, requiring minimal technological know-how and offering the potential for a train-the-trainers approach to maintaining local knowledge. In order to automatically extract parcel boundaries from hand-drawn sketches, several techniques were tested. Blurring as a preprocessing technique gave slight improvements, and Douglas–Peucker generally outperformed convex hulls for vectorizing. The machine learning models did not consistently perform as well as the proposed algorithm for the extraction of the boundaries. The identification of labels from hand-drawn sketches was achieved by innovative character recognition modules automatically reading printed number labels placed on paper maps. All in all, the approach is most suitable for areas where inventory work will

help collect basic information about land tenure/use but is less suitable for areas where the cadastre is already established at a satisfactory level.

**Author Contributions:** Conceptualization, C.L., A.D. and K.K.; method, C.L. and A.D.; participatory mapping, C.L. and K.K.; software development, A.D. and G.V.; software tests, G.V.; writing—original draft preparation, C.L. and A.D.; writing—review and editing, G.V., K.K. and A.S.; funding acquisition, C.L., A.D., K.K. and A.S. All authors have read and agreed to the published version of the manuscript.

**Funding:** This research was funded through the European Social Fund and the Ministry of Economic Affairs, Innovation, Digitalization, and Energy of the State of North Rhine-Westphalia (EFRE-0400389).

**Institutional Review Board Statement:** The study was conducted in accordance with the Declaration of Helsinki, and approved by the Institutional Review Board (or Ethics Committee) of the Institute for Geoinformatics, University of Münster.

**Informed Consent Statement:** Informed consent was obtained from participants involved in the data collection process.

**Data Availability Statement:** The dataset used to train and fine-tune the deep learning models are available at https://doi.org/10.34740/KAGGLE/DSV/6295751. The models and the results are available at https://huggingface.co/aurioldegbelo/slm-unet-080823 and https://huggingface.co/aurioldegbelo/slm-segformer-080823-b1 respectively. The algorithms for boundary extraction using edge detection are available at https://huggingface.co/aurioldegbelo/slm-edge-detection.

**Acknowledgments:** We thank all participants of the different participatory mapping sessions and all fieldworkers involved for their precious contributions. The mapping sessions were done in cooperation with the Youth Initiative for Land in Africa (Yilaa) and Kadaster International. Their support is gratefully acknowledged. We also thank Benjamin Risse, Claudio Persello and Malumbo Chipofya for their helpful feedback about the digitization ideas at the various stages of the project.

**Conflicts of Interest:** The authors declare no conflict of interest. The funders had no role in the design of the study; in the collection, analyses, or interpretation of data; in the writing of the manuscript; or in the decision to publish the results.

## Notes

[1] An important aspect of treating the data responsibly is *reporting*: Since the data collected is sensitive, we make sure in the reporting of the results (e.g., this article) not to display them on a digital map, as they can provide the geographic context of digitized data (and potentially lead to the identification of some participants).

[2] https://getodk.org/, accessed on 31 July 2023.

[3] https://www.digitalocean.com/, accessed on 31 July 2023.

[4] https://www.opendronemap.org/webodm/, accessed on 31 July 2023.

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
