# Peer review of "The SmartLandMaps Approach for Participatory Land Rights Mapping"

_land, doi:10.3390/land12112043_

Round 1

Reviewer 1 Report

Comments and Suggestions for Authors

Dear Authors,

I fully enjoyed reading your manuscript. With my more than 10 years of professional experience working in a cadastral organisation of developing world (we had used plane tabling techniques and chain/tape those days), this article brings me very close to where I had worked before. This article is well written and I could not find any blunder. Being a land administration scholar, I have a few suggestions to improve the quality of this article. I have also attached the marked file where you can see what changes you may need to improve the quality.  I would also like to appreciate you for taking care for formatting and English grammar. The written language is simple to understand as well. I wish your team all the best.

Abstract:

I would suggest you to revise this abstract section adding some more background information at the beginning discussing why this study is important and what is the knowledge gap?

Discuss at the end what contribution this paper is going to provide on theory and practice. It will add value and make the reader aware about the scholarly works it has been done as part of this research study.

Introduction:

Please check the Fit for Purpose Land Administration Publication.

https://www.mdpi.com/books/bookset/4637

There are even 2022 and 2023 quality publications in this area which you can refer/cite here. You can specify which publications/previous works are related to your paper.

Suggest to include one paragraph discussing about the structure of this paper such as Section 1 describes… etc..

Methods:

One of the important activities of participatory land rights mapping is distribution of land/land right ownership certificate to the owner as well. It would be worth to consider that activity as well so that the stakeholders and local authority involved on this activity will feel empowered and motivated for the success of this project. There are many projects in the developing world, the mapping has been completed and the maps and land ownership certificates are in the cabinet. The owners are not able to get the land ownership certificate or parcel certificate.

Involvement of local authority representation or community leader is also important as part of this process. Please specify.

At the Section 3.3, Adjudication is also important before mapping.

Conclusions: Please see my comments. This section needs to be expanded.

References:

Check all the references and be consistent.

For specific comments, please see this attached file (highlighted). 

Reviewer 2 Report

Comments and Suggestions for Authors

The article presents a sound study of how to collect information about the boundaries of land plots and their owners. It should be noted that the application can be well applied in areas where inventory work will help collect basic information about properties. However, it is not suitable for areas where the cadastre is already established at a satisfactory level and the work would significantly improve the accuracy of the information collected.

Reviewer 3 Report

Comments and Suggestions for Authors

Dear authors,

I would like to commend you for an interesting topic with significant practical impact.

Since there is a significant amount of research dealing with the use of GeoAI technologies in the field of parcel/object boundary detection, it would be good to expand the related works section and provide a more detailed analysis (table of analysis) of the improvement of your model and procedures compared to existing research.
